# Effect of a Food for Special Medical Purposes for Muscle Recovery, Consisting of Arginine, Glutamine and Beta-Hydroxy-Beta-Methylbutyrate on Body Composition and Skin Health in Overweight and Obese Class I Sedentary Postmenopausal Women

**DOI:** 10.3390/nu13030975

**Published:** 2021-03-17

**Authors:** Mariangela Rondanelli, Mara Nichetti, Gabriella Peroni, Maurizio Naso, Milena Anna Faliva, Giancarlo Iannello, Enrica Di Paolo, Simone Perna

**Affiliations:** 1IRCCS Mondino Foundation, 27100 Pavia, Italy; mariangela.rondanelli@unipv.it; 2Department of Public Health, Experimental and Forensic Medicine, University of Pavia, 27100 Pavia, Italy; 3Endocrinology and Nutrition Unit, Azienda di Servizi alla Persona “Istituto Santa Margherita”, University of Pavia, 27100 Pavia, Italy; dietista.mara.nichetti@gmail.com (M.N.); mau.na.mn@gmail.com (M.N.); milena.faliva@gmail.com (M.A.F.); 4General Management, Azienda di Servizi alla Persona “Istituto Santa Margherita”, 27100 Pavia, Italy; direttoregenerale@asppavia.it; 5General Geriatric Unit, Azienda di Servizi alla Persona “Istituto Santa Margherita”, 27100 Pavia, Italy; enrica_dipaolo@asppavia.it; 6Department of Biology, Sakhir Campus, College of Science, University of Bahrain, 32038 Sakhir, Bahrain; simoneperna@hotmail.it

**Keywords:** obesity, arginine, glutamine, beta-hydroxy-beta-methylbutyrate (HMB), skin

## Abstract

The consumption of dietary amino acids has been evaluated for therapeutic and safety intervention in obesity. In particular, three molecules have been shown to be effective: arginine, glutamine and leucine (and its metabolite beta-hydroxy-beta-methylbutyrate, HMB). This randomized, double-blinded pilot study in obese postmenopausal patients aimed to evaluate the efficacy of the administration of a specific food for special medical purposes (FSMP) consisting of arginine, glutamine and HMB on body composition, in particular, visceral adipose tissue (VAT), assessed by dual-energy X-ray absorptiometry (DXA), as the primary endpoint. The secondary endpoint was to evaluate the effects on skin health through a validated self-reported questionnaire. A significant improvement on VAT of Δ = −153.600, *p* = 0.01 was recorded in the intervention group. Skin health showed a significant improvement in the treatment group for the following: bright Δ = 1.400 (0.758; 2.042), elasticity Δ = 0.900 (0.239; 1.561), wrinkles Δ = 0.800 (0.276; 1.324), and on total score, Δ = 3.000 (1.871; 4.129). In the intervention group, the improvement in VAT was associated with an improvement in the bright score (r = −0.58; *p* = 0.01). In conclusion, this study demonstrated that the intake for 4-weeks of arginine, glutamine and HMB effects a significant reduction in VAT and improves skin condition, while fat free mass (FFM) is maintained, thus achieving “high-quality” weight loss.

## 1. Introduction

Obesity is defined by a Body Mass Index (BMI) greater than 30 kg/m^2^ and by an imbalance between energy intake/energy expenditure, which can lead to an increased risk of developing insulin resistance and diabetes, and is a major public health concern [1].

Diet and lifestyle changes represent the first line interventions for the body weight management; however, both short and long-term compliance is difficult. Recently, the popularity of dietary supplements (DS) for weight management has increased, and a wide variety of these products are available over the counter [2]. The use of DS, in association with prudent diet and lifestyle, can be a useful aid to improve compliance. In particular, to improve compliance, DS in combination with diet and lifestyle should achieve three objectives: (1) increase the sense of satiety [3]; (2) reduce the loss of fat free mass (FFM) (preserving FFM while losing fat mass (FM), and in particular visceral adipose tissue (VAT), is optimal because It does not affect the basal metabolic rate the muscle strength. In addition It could decrease the negative effect of the visceral adipose tissue on cardiovascular risk and lifespan [4]; (3) minimize the negative aesthetic effects of weight loss on the skin (elasticity, dullness, deepening of wrinkles), which occur because of damage to collagen and elastin, which allows for no skin retraction after weight loss [5]. Among DS that meet the above three objectives, interest has sparked in the use of amino acids in obese subjects in recent years [6,7]. In this regard, the use of dietary amino acids (AA) has been extensively explored for therapeutic and safety intervention of obesity and obesity-induced dysfunction [8]. Paddon-Jones et al. demonstrated that the consumption of essential AA stimulated muscle protein synthesis in the young and elderly people [9]. In particular, 3 AA have been shown to be effective both in animal models and in humans in promoting the loss of fat mass, specifically visceral adipose tissue, and in maintaining free fat mass: arginine, glutamine and leucine (or its metabolite beta-hydroxy-beta methylbutyrate).

Considering arginine, this amino acid protects rats fed a high-fat diet from an increase of visceral adipose tissue, possibly through a change in MMP-2 and MMP-9 activity and amelioration of insulin sensitivity [10]. In humans, 3-months L-arginine supplementation (9 g/day) improves insulin sensitivity in patients with visceral obesity [11]. Moreover, increase of blood growth hormone was also demonstrated after L-arginine intake [12].

The AA glutamine is the precursor of L-arginine, which generates nitric oxide (NO) [13] and glutathione, a major biological antioxidant [14].

The metabolite of leucine, β-hydroxy-β-methylbutyrate (HMB), was found to improve the metabolic ability to use fat and increase oxidation of fatty acids in adipocytes and muscle cells in an animal model study conducted by Bruckbauer [15]; it also determines an increase in insulin sensitivity, reduction of insulin resistance and attenuation of systemic inflammatory stress, as reported in a recent review by He [16].

In humans, some studies show that using HMB as a supplement during physical activity or in low-calorie diets can be effective in maintaining lean body mass [17].

HMB in combination with endurance exercise resulted in lower whole body fat values [18,19,20] and a reduction in fat mass in elderly adult men [21]. Moreover, in a study [22] where the effect of HMB supplementation for 6 weeks on muscle strength and body composition in sedentary and overweight women was considered, a reduction in weight, waist and abdominal circumference has been demonstrated, as well as an increase in strength, even without resistance training and without changes in lean mass.

A recent study by Takaoka demonstrated that in young adult women without any fitness habit, body fat tended to decrease, while the amount of muscle increased after 4 weeks supplementation with these three amino acids [23]. Another interesting and innovative result of the study by Takaoka is the improvement in skin texture. Recently, the scientific literature has begun to consider the criticality concerning the skin changes that occur after following a low-calorie diet with weight loss. Arginine, glutamine and HMB are involved in maintaining healthy skin, as they stimulate the synthesis and deposition of collagen [24,25,26,27]. A study by Williams demonstrated that collagen synthesis has been significantly enhanced in healthy elderly volunteers by the oral administration of a mixture of arginine, HMB and glutamine [28].

Given this background, the aim of this randomized, double-blinded pilot study was to evaluate the efficacy of a specific food for special medical purposes (FSMP) consisting of arginine, glutamine and HMB on body composition (Fat Mass—FM and Fat Free Mass—FFM), and in particular, visceral adipose tissue (VAT) reduction, as the primary end point in overweight and obese class I sedentary postmenopausal women. The secondary end point is the evaluation of skin condition by a validate self-reported questionnaire survey after 4-weeks intake of this FSMP. Overweight and obese class I sedentary postmenopausal women [29,30].

## 2. Materials and Methods

### 2.1. Study Disegn

This was a randomized, double-blinded pilot study.

### 2.2. Study Population

The study enrolled obese class I and overweight (BMI 25–35 kg/m^2^) postmenopausal women admitted, as outpatients, to the Dietetic and Metabolic Unit of the “Santa Margherita” Institute, University of Pavia, Pavia, Italy. Ethics Committee of the University of Pavia (ethical code number: 0912/14122018, registered with ClinicalTrials.gov: NCT04701463).

Inclusion criteria: Subjected aged: from 45 to 65 years enrolled between January 2020 and November 2020. The subjects were free of overt renal, liver and thyroid disease and were not taking any medication, with no cigarette and alcohol consumption over.two standard alcoholic beverages/day (20 g of alcohol/day) Physical activity was recorded and sedentary subjects, as defined by Sedentary Behavior Research Network [31], were admitted to the study.

The experimental protocol was approved by the Local Ethics Committee of the University of Pavia (ethical code number: 0912/14122018). The study was registered with ClinicalTrials.gov: NCT04701463. All the volunteers signed a written informed consent.

### 2.3. Dietary Supplement Intervention

The dietary supplement intervention was given as 2 daily servings (mid-morning and mid-afternoon) of an oral food for special medical purposes (Abound^®^, Abbott, USA) consisting of: L-glutamine, L-arginine, acidity regulator: E330, calcium beta-hydroxy-beta-methylbutyrate (Ca-HMB), powdered orange juice, sucrose, turmeric powder, sweeteners (E951, E950), MCTs from palm seed oil and coconut oil in varying percentages, color E162, or an isocaloric placebo. The supplementation period was 4 weeks. The dietary supplements were delivered at the time of the first blood sample.

The maltodextrin placebo assigned to the control group was an isocaloric formula with the same appearance as the study product.

Subjects were randomized to (in a 1:1 ratio).

### 2.4. Adverse Events

Adverse events were assessed and recorded with participant self-reports. At the start and at the end of supplementation also safety blood biochemistry parameters for renal and liver function were evaluated.

### 2.5. Biochemical Markers

Serum insulin, Fasting blood glucose (FBG), total cholesterol (TC), low-density lipoprotein-cholesterol (LDL-C), high-density lipoprotein-cholesterol (HDL-C) and tri-glyceride (TG) levels, Homeostasis Model Assessment (HOMA) [32]. were measured at baseline (t0), and after 30 days, at the end of treatment (t1).

### 2.6. Anthropometric Measurements and Dietary Advice

Anthropometric measurements were assessed at baseline (t0) and after 30 days, at the end of supplementation (t1). Body weight and height were measured following a standardized technique [33] and BMI was calculated (kg/m^2^). Height was determined using a height meter with an accuracy of 1 mm (range, 80–200 cm). Body weight was measured and BMI was calculated as body weight ((kg)/(height (m)^2^). 

Participants were recommended to follow a diet restricting their daily energy intake by 500 kcal/d less than their daily requirements based on WHO criteria (World Health Organization, 1985, Washington, DC, USA)

### 2.7. Body Composition

Body composition FFM, FM, VAT volumewas measured by dual-energy X-ray absorptiometry (DXA) (Lunar Prodigy, GE Medical Systems) at baseline (t0) and after 30 days, at the end of supplementation (t1) [34].

### 2.8. Assessment of Skin Condition

The participants were trained to complete a questionnaire survey with a five-point evaluation scale to assess the skin facial condition (luster, suppleness, and wrinkles) [35] at baseline and after 4 weeks, at the end of the study.

### 2.9. Primary and Secondary Endpoints

We evaluated as primary endpoint the assessment of VAT evaluated by DXA. The secondary endpoints were the assessment of skin condition and the other parameters of body composition (FM and FFM).

### 2.10. Sample Size

The sample size was determined based on the previous study by Takaoka et al. [23].

### 2.11. Statistical Analysis

This study follows the CONSORT guidelines for clinical trials [36]. For the baseline variables, frequencies, d mean and standard deviation (SD) for continuous variables have been summarized. Between placebo and treatment group at baseline have beencompared using chi-square test f. In the primary analysis, pre post-adjusted means differences and 95% confidence interval (CI) estimated by analysis of covariance with the change in the primary outcome (the skin markers) at 4 weeks were compared between the intervention group and control. Univariate analysis was applied for the secondary outcomes (body composition) at each time point (4 weeks). *p* value < 0.05 was considered statistically significant. All analyses were performed by SPSS 21software (IBM, Chicago, USA).

## 3. Results

A total of 20 females were randomly assigned to the supplement group (10) or placebo group (10). The baseline characteristics of the subjects studied are shown in Table 1.

Figure 1 shows the flow chart of the study.

There was no statistically significant difference between the two groups at baseline for each parameter. Among the 20 subjects included in the analysis, the mean age was 54.44 ± 5.33 years, and the mean BMI (±SD) was 29.51 ± 2.90 kg/m^2^. All parameters were within the normal range according to WHO criteria.

As shown in Table 2, the decrease in body weight −1.540 kg (−2,484; −0.596), BMI −0.597 kg/m^2^ (−0.977; −0.217), VAT −93.800 g (−140.650; −46.950) and FM −1,166.300 g (−2028.800; −303.800) were statistically significant in the treatment group (*p* = 0.01). No positive effect was recorded in the placebo group. Between-group analysis showed an improvement (statistically significant) on VAT of Δ = −153.600, *p* = 0.01 (treatment minus placebo).

As shown in Table 3, the main markers for skin effects were statistically significantly improved in intervention group. The oral supplement group reported an improvement in brightness Δ = 1.400 (0.758; 2.042), elasticity Δ = 0.900 (0.239; 1.561), wrinkles Δ = 0.800 (0.276; 1.324) and on total score Δ = 3.000 (1,871; 4.129). No statistically significant improvement was recorded in the placebo group. Between-group analysis showed an improvement in all main markers (statistically significant) (treatment minus placebo effects) brightness Δ = 1.400, elasticity Δ = 0.900, wrinkles Δ = 0.800 and on total score Δ = 3.100.

For the Estimated Marginal Means for treatment, there was a statistically significant decrease in VAT of −93.800 g (−140.650; −46.950) in the intervention group and an increase of 59.800 g (12.950; 106.650) in the placebo group. The effect between groups was statistically significant. 

The between-group effects of treatment (treatment minus placebo) showed an improvement for brightness from 0.685 to 2.114, for elasticity from 0.169 to 1.631 and for wrinkles from 0.335 to 1.465.

As shown in Figure 2, the improvement in skin markers was associated with a high level of coefficient of correlation with a decrease in Δ VAT and increase in Δ brightness = −0.78. A similar association was recorded for waist circumference and brightness (r = 0.63).

Finally, Figure 3 shows the delta change in correlation split by groups (intervention and placebo) between VAT and total score of the five-point evaluation scale. No correlation was recorded in the placebo group, while the improvement of Δ change was associated with a reduction of VAT in intervention group (r = −0.58; *p* = 0.01).

None of the intervention or control group refused the supplement or placebo and no side effects were reported during the study.

## 4. Discussion

The principal finding of this study is that consumption of an oral nutrition supplement containing arginine, glutamine and HMB for 4 weeks in sedentary postmenopausal women demonstrates a statistically significant reduction both intra-group and inter-group of the primary endpoint VAT, and secondary endpoints, such as body fat, while fat free mass is maintained. Hence it can be said that the goal of “high-quality” weight loss has been achieved, since FFM is closely related to lifespan [4].

In addition, a decrease in VAT is very important for high-quality weight loss because a reduction in VAT leads to a reduction in cardiovascular risk [37], a very important consideration for postmenopausal women [38].

The results of our study are in agreement with the results of the study by Takaoka, which demonstrated that in subjects with similar characteristics, namely, young, sedentary women, body fat tended to decrease, while the amount of body muscle increased after 4 weeks supplemental intake of these three amino acids, though the nutrient levels in the FSMP are different (L-leucine 600 mg, L-arginine 250 mg, and L-glutamine 300 mg in the Takaoka study and HMB, metabolite of leucine, 1300 mg, arginine 7400 mg and glutamine 7400 mg in our study) [23]. This result is due to the synergy of activity of arginine, glutamine and HMB, a metabolite of leucine, as all three of these molecules have a demonstrated stimulating action on protein synthesis and lipolysis.

Recent studies in animal models showed that HMB improves metabolic capacity to utilize fat and increases fatty acid oxidation in adipocytes and muscle cells [15,16], and in humans, HMB in combination with resistance exercise resulted in lower values for whole body fat [18,19,20] and decreased adipose fat mass in older adult men [21,22].

Moreover, growth hormone, a lipolytic hormone, is increased after HMB and arginine intake [12,21,39].

Finally, arginine protects rats fed a high-fat diet from an increase in visceral fat, possibly through a change in MMP-2 and MMP-9 activity and amelioration of insulin sensitivity [10] and also in humans, 3-months L-arginine supplementation improves insulin sensitivity [11].

Another surprising result of this study is on skin condition; the consumption of the FSMP by sedentary postmenopausal women for 4 weeks led to a statistically significant amelioration both intra group and inter group on the brightness, suppleness, and wrinkles of the facial skin. This effect on the skin is also similar between our study and the study by Takaoka. The etiology of skin changes after a hypocaloric diet with weight loss is still inadequately understood today. It usually occurs because of damage to collagen and elastin, which does not allow for skin retraction after weight loss [5,40,41,42,43,44].

These findings were attributable to the additive effects produced by the combination of arginine, glutamine and HMB because all these three molecules stimulate the synthesis and deposition of collagen, both individually [24,25,26] and together, as demonstrated in the study by Williams in which it is reported that collagen synthesis has been significantly enhanced in healthy elderly volunteers by the oral administration of a mixture of arginine, HMB and glutamine [28]. Finally, in addition to the effect on the synthesis and deposition of collagen, it is also important to underline the role of HMB in the metabolism of cholesterol, which is a fundamental lipid of the epidermal barrier [45].

Through successive steps, HMB is converted within the cytoplasmic matrix of muscle cells into beta hydroxy beta methylglutaryl coenzyme A (HMG-CoA). HMG-CoA is converted into cholesterol. The muscle independently produces its own cholesterol, typically from HMG-CoA, to preserve the integrity of cell membranes, since it is unable to meet its own needs through absorption from the circulation [46].

Of note is also the statistically significant positive correlation found between VAT and the individual scores and the total score of the self-reported questionnaire on facial skin condition. This result was demonstrated for the first time in the literature. However, the innovation of the study consists precisely in having investigated this new interesting aspect concerning the relationship between weight loss, body composition and skin health. On this innovative topic, given the lack of other papers published in the literature, we can therefore only formulate hypotheses.

Visceral adipose tissue causes increased angiogenesis, immune cell infiltration, the overproduction of extracellular matrix and the increased production of pro-inflammatory adipocytokines [47]. A reduction of VAT and therefore of inflammation can have an improving action on the quality of the skin.

Several limitations of this study must be discussed. The main limitation is that there are no placebo arms for the single molecules, which would have enabled the role of the molecules in combination to be clearly demonstrated. Moreover, since the sample size was small and only overweight and obese class I postmenopausal women were enrolled, all these findings must be interpreted with caution, and further studies are needed with a larger sample size from the general population

## 5. Conclusions

In conclusion, the results of the study demonstrated that consumption for 4-weeks of an oral supplement containing arginine, glutamine and HMB in combination demonstrates, in sedentary postmenopausal women, a statistically significant reduction both intra group and inter group of the primary endpoint VAT and the secondary endpoints, such as body fat, while fat free mass is maintained. Thus, the goal of “high-quality” weight loss has been achieved, since FFM is closely related to basal metabolic rate, functional capacity and lifespan. Moreover, the skin condition is significantly improved.

## Figures and Tables

**Figure 1 nutrients-13-00975-f001:**
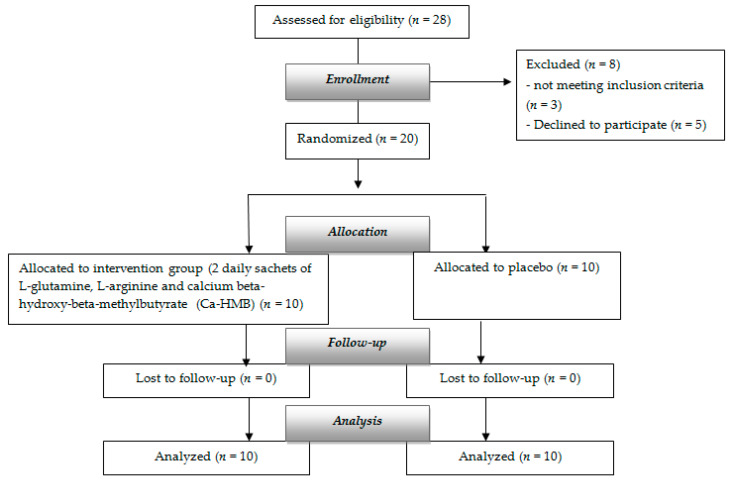
Flow chart of the study.

**Figure 2 nutrients-13-00975-f002:**
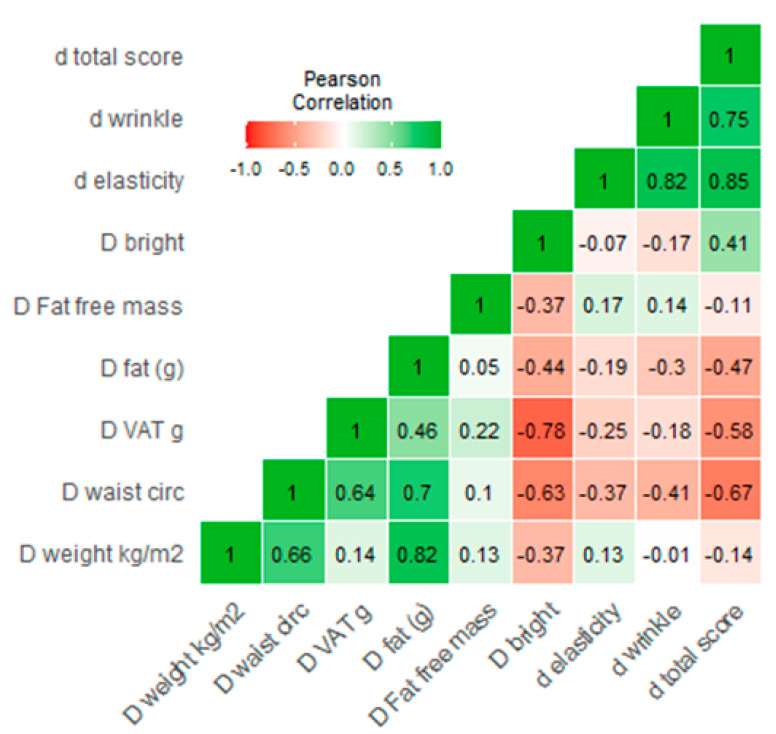
Correlation heatmap showed the association between the delta changes for intervention group.

**Figure 3 nutrients-13-00975-f003:**
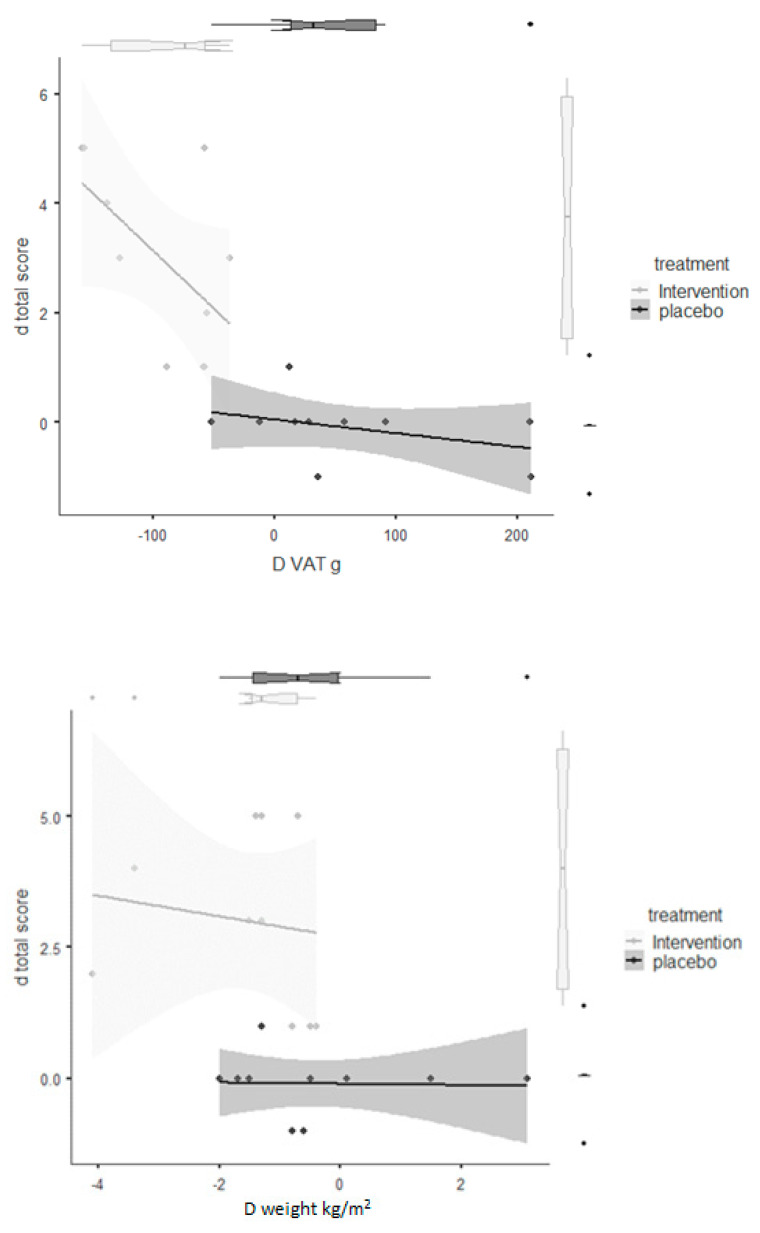
Delta change correlation divided by groups (intervention and placebo) between VAT and total score of five-point evaluation scale.

**Table 1 nutrients-13-00975-t001:** Clinical characteristics at baseline.

Outcome	Intervention Group (*n* = 10) Mean ± SD	Placebo Group (*n* = 10) Mean ± SD	Total Sample (*n* = 20) Mean ± SD	*p*-Value between Groups
**Age (years)**	54.60 ± 5.64	54.20 ± 5.29	54.44 ± 5.33	0.872
**Total Cholesterol (mg/dL)**	201.70 ± 23.45	198.80 ± 40.07	200.25 ± 31.76	0.846
**LDL Cholesterol (mg/dL)**	119.70 ± 10.40	121.44 ± 34.07	120.57 ± 22.24	0.879
**HDL Cholesterol (mg/dL)**	66.90 ± 15.39	64.70 ± 12.50	65.80 ± 13.95	0.730
**Triglycerides (mg/dL)**	100.80 ± 28.07	99.60 ± 25.24	100.20 ± 26.66	0.921
**Glycemia (mg/dL)**	96.70 ± 8.47	96.00 ± 12.71	96.35 ± 10.59	0.886
**Insulin (mcIU/mL)**	11.77 ± 4.49	11.42 ± 4.11	11.60 ± 4.30	0.859
**HOMA index (pt)**	2.85 ± 1.25	2.79 ± 1.30	2.82 ± 1.28	0.922
**Aspartate Aminotransferase (UI/L)**	20.50 ± 4.17	19.10 ± 3.21	19.80 ± 3.69	0.411
**Alanine Aminotransferase (UI/L)**	22.70 ± 7.09	20.20 ± 5.18	21.45 ± 6.14	0.380
**Gamma Glutamyl Transferase (U/L)**	26.10 ± 7.50	23.00 ± 7.99	24.55 ± 7.75	0.383
**Creatinine (mg/dL)**	0.70 ± 0.15	0.73 ± 0.06	0.72 ± 0.11	0.515
**eGFR mL/min**	90.37 ± 15.07	88.90 ± 9.90	89.64 ± 12.49	0.800
**Height (m)**	1.60 ± 0.05	1.63 ± 0.05	1.62 ± 0.05	0.272
**Weight (kg)**	74.72 ± 6.54	78.69 ± 9.86	76.71 ± 8.20	0.303
**Body Mass Index (kg/m^2^)**	29.20 ± 2.78	29.82 ± 3.01	29.51 ± 2.90	0.635
**Waist circumference (cm)**	95.45 ± 5.96	99.00 ± 9.61	97.23 ± 7.79	0.334
**Visceral Adipose Tissue (g)**	984.30 ± 398.71	1119.20 ± 455.96	1051.75 ± 427.34	0.490
**Fat Mass (g)**	32,419.40 ± 5642.07	32,044.90 ± 6251.10	32,232.15 ± 5946.59	0.890
**Fat Free Mass (g)**	40,443.40 ± 2412.14	44,010.50 ± 7942.71	42,226.95 ± 5177.42	0.191
**Bright (pt)**	2.60 ± 0.97	2.60 ± 0.97	2.60 ± 0.97	1.000
**Elasticity (pt)**	2.70 ± 0.95	2.80 ± 0.63	2.75 ± 0.79	0.785
**Wrinkle (pt)**	2.90 ± 0.57	2.90 ± 0.74	2.90 ± 0.66	1.000

Abbreviations—eGFR: Glomerular Filtration Rate, HDL: High Density Lipoprotein, LDL: Low Density Lipoprotein, SD: standard deviation.

**Table 2 nutrients-13-00975-t002:** Within-group and between-group differences for anthropometric measurements and body composition for the intervention and placebo groups. The estimate of the effect β, its 95% confidence interval (CI) and the adjusted *p*-value of the null hypothesis of a null effect are reported.

Variable	Intervention Intra-Group β (95%CI)	Placebo Intra-Group β (95%CI)	Intervention Effect between Groups Mean Difference
**Weight (kg)**	**−1.540 (−2.484; −0.596)**	−0.370 (−1.314; 0.574)	−1.170
**Body Mass Index (kg/m^2^)**	**−0.597 (−0.977; −0.217)**	−0.080 (−0.460; 0.300)	−0.517
**Waist circumference (cm)**	−1.550 (−3.240; 0.140)	−0.050 (−1.740; 1.640)	−1.500
**Visceral Adipose Tissue (g)**	**−93.800 (−140.650; −46.950)**	**59.800 (12.950; 106.650)**	**−153.600**
**Fat Mass (g)**	**−1166.300 (−2028.800; −303.800)**	225.100 (−637.400; 1087.600)	**−1391**
**Fat Free Mass (g)**	−230.500 (−915.600; 454.600)	−576.200 (−1261.300; 108.900)	345.600

In bold: value with *p* < 0.05.

**Table 3 nutrients-13-00975-t003:** Within-group and between-group differences for skin effects for the intervention and placebo groups. The estimate of the effect β, its 95% confidence interval (CI) and the adjusted *p*-value of the null hypothesis of a null effect are reported.

Variable	Intervention Intra-Group β (95%CI)	Placebo Intra-Group β (95%CI)	Intervention Effect between Groups Mean difference
**Bright**	**1.400 (0.758; 2.042)**	0.000 (−0.313; 0.313)	1.400
**Elasticity**	**0.900 (0.239; 1.561)**	0.000 (−0.313; 0.313)	0.900
**Wrinkles**	**0.800 (0.276; 1.324)**	−0.100 (−0.310; 0.110)	0.900
**Total score**	**3.000 (1.871; 4.129)**	−0.100 (−0.477; 0.277)	3.100

In bold: value with *p* < 0.05.

## Data Availability

The data presented in this study are available in the article.

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
