# Peer review of "Effect of a Food for Special Medical Purposes for Muscle Recovery, Consisting of Arginine, Glutamine and Beta-Hydroxy-Beta-Methylbutyrate on Body Composition and Skin Health in Overweight and Obese Class I Sedentary Postmenopausal Women"

_nutrients, 2021, doi:10.3390/nu13030975_

Round 1

Reviewer 1 Report

This study evaluated the efficacy of Abound® on the body composition and skin health. The authors showed that one of FSMPs including special amino acids may help improvement of the outcomes.

  1. What is the definition of sedentarism? The definition was not clearly given.
  2. The authors provided the participants with Abound or placebo. Is there more intervention protocol for them? Generally, Dietetic and Metabolic Units may instruct the ways of diet and exercise.
  3. How did this study decide who was taking IP or placebo? Is this double-blind study? Please show randomization method.
  4. The questionnaires for skin health may be subjective. In addition, why the authors think that the FSMP promote the skin health? Further description should be required for this point.
  5. Figure 1 and 2 does not provide additional information to Table 2 and 3. They are redundant.

Reviewer 2 Report

I was honored to review the manuscript entitled “Effect of a food for special medical purposes for muscle recovery, consisting of arginine, glutamine and beta-hydroxy-beta-methylbutyrate on body composition and on skin health in overweight and obese class I sedentary postmenopausal women.” submitted to Nutrients. The study presents high quality and deals with important clinical issue, such type of study is needed.  I have only few small remarks that authors should address properly.

There are only some points to correct:

 - please provide the list of abbreviations

 - please provide the number of ethical approval

- introduction and discussion section need improvement – please provide information on how your results will translate into clinical practice

- in discussion section please provide study strong points  and study limitation section

- please correct typos

Reviewer 3 Report

STRUCTURE

  • The manuscript is properly structured

TITLE AND ABSTRACT

  • The title or abstract should inform that the type of study
  • The title of your manuscript should be concise

INTRODUCTION

  • Scientific background and explanation of rationale is limited
  • Explain more about the study population in the introduction
  • State specific objectives
  • The hypotheses are missing

MATERIAL AND METHODS

  • Present key elements of study design early in the paper
  • Describe the settings and locations where the data were collected
  • How many people participated in the study?
  • Only obesity according to BMI was taken into account? Was it considered to be assessed according to % body fat?
  • What instruments were used to weigh and measure height?
  • Were weight and height measurements taken early in the day? All at the same time?
  • Was the skin measurement questionnaire validated?
  • Include how and when the supplements were actually administered
  • How many people were in each group?
  • Was the distribution between intervention and control group randomised?
  • What was the method used to generate the random allocation sequence
  • Define clearly all exposures, predictors, potential confounders, and effect modifiers
  • Who generated the random allocation sequence, who enrolled participants, and who assigned participants to interventions?
  • Who was blinded after assignment to interventions? And how?
  • Were there any changes to trial outcomes after the trial commenced?

RESULTS

  • Consider to use a flow diagram
  • Dates defining the periods of recruitment and follow-up
  • Why the trial ended or was stopped?
  • If the table is cut into two different sheets, reinstate the header of the table
  • Add all important harms or unintended effects
  • The quality of Figure 1 needs to be improved
  • The quality of Figure 2 needs to be improved
  • Explain all the abbreviations in Figure 3 and the quality must be improved
  • The quality of Figure 4 needs to be improved

DISCUSSION

  • Add reference at line 266
  • The discussion needs to go deeper.
  • Give a cautious overall interpretation of results considering objectives, limitations, multiplicity of analyses, results from similar studies, and other relevant evidence
  • Discuss the generalisability (external validity) of the study results

REFERENCES

  • Some of the references do not follow the indicated style, please review all of them. 

Round 2

Reviewer 3 Report

Please, explain how was randomized
